# Diagnostic, Prognostic and Mechanistic Biomarkers of COVID-19 Identified by Mass Spectrometric Metabolomics

**DOI:** 10.3390/metabo13030342

**Published:** 2023-02-24

**Authors:** Mélanie Bourgin, Sylvère Durand, Guido Kroemer

**Affiliations:** 1Metabolomics and Cell Biology Platforms, Institut Gustave Roussy, 94805 Villejuif, France; 2Centre de Recherche des Cordeliers, Equipe Labellisée par la Ligue Contre le Cancer, Université de Paris Cité, Sorbonne Université, Inserm U1138, Institut Universitaire de France, 75005 Paris, France; 3Institut du Cancer Paris CARPEM, Department of Biology, Hôpital Européen Georges Pompidou, AP-HP, 75610 Paris, France

**Keywords:** COVID-19, metabolomics, prognostic, diagnosis, biomarkers, mass-spectrometry

## Abstract

A number of studies have assessed the impact of SARS-CoV-2 infection and COVID-19 severity on the metabolome of exhaled air, saliva, plasma, and urine to identify diagnostic and prognostic biomarkers. In spite of the richness of the literature, there is no consensus about the utility of metabolomic analyses for the management of COVID-19, calling for a critical assessment of the literature. We identified mass spectrometric metabolomic studies on specimens from SARS-CoV2-infected patients and subjected them to a cross-study comparison. We compared the clinical design, technical aspects, and statistical analyses of published studies with the purpose to identify the most relevant biomarkers. Several among the metabolites that are under- or overrepresented in the plasma from patients with COVID-19 may directly contribute to excessive inflammatory reactions and deficient immune control of SARS-CoV2, hence unraveling important mechanistic connections between whole-body metabolism and the course of the disease. Altogether, it appears that mass spectrometric approaches have a high potential for biomarker discovery, especially if they are subjected to methodological standardization.

## 1. Introduction

The pandemic outbreak of coronavirus disease 2019 (COVID-19) has caused a major perturbation of public health, coupled to global social and environmental change. As of 8 February 2023, the World Human Organization (WHO) reported 754 million confirmed cases of COVID-19, including 6.8 million deaths [1]. The causative agent of COVID-19 is severe acute respiratory syndrome coronavirus-2 (SARS-CoV-2), which potentially affects the entire human body, impacting the respiratory, inflammatory, neurological, cardiovascular, and gastrointestinal systems [2]. Most individuals infected with SARS-CoV-2 develop polymorphic symptoms ranging from close-to-asymptomatic to severe clinical illness requiring hospitalization and sometimes even multiorgan failure leading to death [3,4]. Nowadays, novel SARS-CoV-2 strains variants still evolve in the unfolding COVID-19 pandemic [5]. Aged and immunocompromised persons, as well as obese individuals with co-morbidities (such as diabetes and hypertension), are particularly susceptible to SARS-CoV-2 infection and the development of COVID-19 [1,2,4,5,6,7,8]. Traditional biomolecular diagnostic tests, particularly the detection of viral DNA by reverse transcriptase real-time PCR (RT-qPCR) in oro-nasopharyngeal swabs, have played a significant role in tracking SARS-CoV-2 [9]. Alternatively, antigen detection immunoassays identify specific viral proteins in nasopharyngeal swabs or saliva specimens [4,8].

In the fight against the COVID-19 pandemic, mass spectrometry (MS)-based metabolomics constitutes one of the cutting-edge technologies allowing to detect and identify circulating metabolites as potential biomarkers of SARS-CoV-2 infection and disease severity [10,11,12]. Furthermore, metabolomic allows to characterize the complex relationship between SARS-CoV-2 and host metabolism, which manifests a rewired TCA cycle, reducing oxidative glutamine metabolism, and increasing pyruvate entry [13,14,15]. Thus, metabolomics provides a systemic dimension of the host–virus interaction [15,16,17]. This approach sheds light on the determinants of biochemical pathways associated to COVID-19 disease from the initial steps of infection to its progression resulting in final resolution or lethal aggravation [11,18,19,20,21,22]. When comparing different metabolomic analyses of COVID-19 specimens, it is noteworthy that there is a big heterogeneity in methods, ranging from (i) clinical test design, to (ii) sample collection and preparation, (iii) chromatographic and MS-based methods, and (iv) statistics and data processing [23,24,25,26,27,28]. In spite of the need for further methodological standardization, we attempted an extensive overview of the literature on the COVID-19-related metabolome.

Of note, variations in the systemic metabolite profile provide a close-to-holistic view of complex adaptive responses and may pinpoint valuable biomarkers of COVID-19. Extensive data support the contribution of metabolic perturbations to the immune-inflammatory alterations that characterize disease progression in COVID-19 [29,30,31]. However, MS-metabolomics generates large datasets on several hundreds to thousand metabolites that are difficult to interpret in statistical terms [11,32]. In addition, the few studies which were conducted to examine metabolite and pathway-based dysfunctions linked to the diagnosis and prognosis of COVID-19 did not target the same metabolites, prohibiting interexperimental comparisons [10]. To identify the best diagnostic and prognostic biomarkers, we applied a funnel method selecting the most significant metabolites identified by MS-based metabolomics analyses in the literature. This method highlights the top-ranked pathways and creates a panel of biomarkers associated with prognosis. We have gathered mechanistic evidence on diagnostic and prognostic biomarkers, especially those related to the immunopathology associated to multiorgan failure and death [32]. We surmise that metabolomics constitutes a powerful approach for the identification of disease-relevant diagnostic and prognostic biomarkers.

## 2. From Metabolomics to Relevant Prognostic Biomarkers of COVID-19

Metabolomics is the systematic determination of small molecules <1000 Da in biofluids, cells and biological tissues. Metabolites can be produced by the host organism, absorbed with the diet, produced by microorganisms, or stem from other exogenous sources, e.g., aerosols and cosmetic products [33,34]. Metabolomics has been widely used for biomarker discovery, with the goal to identify metabolites that correlate with various diseases [34,35]. This also applies to COVID-19 disease following a non-standardized workflow involving a stereotyped sequence of steps (Figure 1).

### 2.1. Metabolomic Workflow

#### 2.1.1. Clinical Design

The clinical definition of health, disease and disease stages is amongst the features that affect metabolomics workflow (Figure 1) [22]. Multiple studies have explored the COVID-19 metabolome based on a variety of criteria indicating disease progression, such as the severity of pathology (i.e., mild versus severe), outcome (i.e., discharge versus decease), risk factors for severe disease (i.e., comorbidities and cancer), longitudinal assessments of individuals with SARS-CoV-2 infection (i.e., from asymptomatic infection to symptomatic disease, resolution or death), and different types of treatment (e.g., ambulatory versus hospitalization) [35,36,37,38,39,40]. These criteria are so heterogeneous as to compromise comparisons among studies. Moreover, systematic bias can be introduced by distinct methods of handling and storing specimens [14]. For example, the comparison between studies dealing with serum versus plasma metabolomes as well as different storage methods (−80 °C, liquid nitrogen) may be problematic [36,41,42,43].Fortunately, to strengthen metabolomic analyses, multiple studies have established a robust clinical strategy. Briefly, clinical standardization collection can be achieved by applying stringent inclusion criteria, such as viral detection by positive RT-PCR (rather than less accurate test such as viral antigen detection), and broad clinical characteristics (e.g., age, pre-admission symptoms, duration or positivity determined by RT-PCR and the presence of comorbidities) [44]. The severity of the COVID-19 needs to be defined to accurately distinguish mild, moderate, and severe disease, as described in detail by Danlos et al. [41]. An important point is also to include technical and biological replicates to assess the reproducibility of measurements [45]. Finally, we strongly recommend the obligatory exploration of validation cohorts to confirm the identification of potential biomarkers [30,39,44] (Figure 1).

#### 2.1.2. Sampling Preparation and Chromatography Techniques

Within biological samples, the instantaneous cessation of metabolism is critical for the obtention of a snapshot of the metabolic state [46]. Most of the literature on the circulating COVID-19 metabolome is currently based on common protocols (cooling, freezing and unfreezing, extraction with solvents) that must be applied to all samples of the same cohort [41,44,47]. Saliva has limited value for metabolic studies due to the contamination of samples with food items and bacterial products [48]. Moreover, swab kits issued from different manufacturers potentially differ in the metabolites they sample and release [18]. Expired breath can be used to detect volatile organic compounds (VOCs), circumventing the need for such kits [49]. Hence, “electronic noses” are being designed to detect disease-relevant VOCs [50,51].

For optimal data quality, the analytical process must adhere to a standardized procedure that, beyond sample preparation, involves (i) randomization of biological samples, (ii) constant quality control, (iii) batch correction throughout the metabolomic experiment, and (iv) monitoring of the performance and stability of the instruments [32]. In both targeted and non-targeted metabolomics, different methods of chromatography are coupled to an array of mass spectrometers. Typical combinations are liquid chromatography coupled to tandem mass spectrometry (LC-MS/MS), gas chromatography–mass spectrometry (GC-MS), and ultra-performance liquid chromatography-tandem mass spectrometry (UPLC-MS/MS), which may involve simple to multi-quadrupole mass analyzers (QMS), triple-quadrupole ion trap (QTrap), triple quadrupole (TQ), or quadrupole-TOF (Q-TOF) [37,38,52]. However, there are hurdles to the chromatographic separation of metabolites. For example, a short-duty analysis by means of a fast solvent gradient is often recommended to achieve high-throughput of samples containing low molecular weight compounds such as VOCs. These include chromatographic parameters, such as successive times during the gas flow ramp and GC temperature, that affect the detection of candidate COVID-19 breath biomarkers [53]. Flow injection MS/MS is the most popular method to investigate nasal mucosal fluids and ultimately offers high-throughput without compromising sensitivity, precision, and accuracy with a dynamic range from parts per billion to parts per million [18]. The untargeted metabolomic approach has boosted the discovery of potential biomarkers in COVID-19 [39,41,54]. Most often, based on UPHLC-MS/MS, this method facilitates the comprehensive and exploratory detection of metabolites which, however, must be identified in a subsequent step by means of a targeted method. Hence, an important goal of sample preparation is to cover a maximum number of metabolites. This is achieved by means of a generic extraction, allowing for the solubilization of chemically diverse molecules present in the sample. With amphiphilic properties, liquid chromatography (LC) solvents and gradient elution profiles can be adjusted for long chain acylated carnitines, fatty acids, or phospholipids [39]. It is useful to compile targeted and untargeted metabolomic approaches to achieve a comprehensive view of the metabolome [30,41].

#### 2.1.3. MS Based Metabolomic Analysis & Data Processing

The principle of mass spectrometry involves the ionization of chemical compounds to generate charged molecules. These are detected and displayed as spectra with the metabolite mass-to-charge ratio (m/z) in the abscissa and the intensity in the ordinate [11]. To compare MS-based metabolomics methods between different laboratories, it is important to understand whether the ionization source covers the whole metabolome, especially with the untargeted approach [32]. A specific ionization source, electron ionization (EI), is used for GC-MS analysis, e.g., to determine exhaled breath metabolomes from COVID-19 patients [49,55]. Other, less reproducible modes of ionization than EI exist, such as electrospray ionization (ESI). The combination of two different ionization modes, ESI (+) and ESI (−), is applied to the detection of specific compounds, such as chain acylcarnitines, phosphatidylcholines (PCs), lysophosphatidylcholines (LPCs). Amino acids are usually detected in the ESI (+) mode, and fatty acids (FAs) derivatives or bile acids in the ESI (−) mode. The use of both ESI modes provides the advantage of abrogating ionization mode bias and allows the detection of multiple distinct metabolites present in COVID-19 patient samples [54]. As mentioned above, the m/z analyzer is a central pillar to establish metabolomic analysis. Diverse mass analyzers, including QTrap, have been employed to unravel COVID-19-related metabolic biomarkers, such as purine nucleosides, phenols, fructose, and mannose compounds, whereas Orbitrap has been used by Thomas et al. to identify alterations in the kynurenine pathway, nitrogen metabolism, amino acids, oxidative stress, circulating glucose levels, and free fatty acids [42,56].

Detection and integration of m/z peaks from the raw intensity data are critical for the success of qualitative and quantitative analyses. Today, the prevalent metabolite identification is based on data acquisition with (i) the m/z value of a molecular ion, (ii) retention times (RT), (iii) detected ion intensities, and other biochemical parameters listed in authenticated reference standard databases (such as Kyoto Encyclopedia of Genes and Genomes, KEGG, and Human Metabolome Database, HMDB) [32,57]. Insufficient or inaccurate annotation of putative metabolites constitutes a persistent obstacle for metabolomic profiling. A key example is provided by deoxy-fructosyl-amino acids, which are new putative plasma biomarkers for SARS-CoV-2 infection and COVID-19 severity and which were not listed in the databases [58]. Faced with one of the major bottlenecks in current metabolomics studies, recent work using machine-learning algorithms has facilitated the identification of unique metabolite biomarkers [36]. Zhu and colleagues developed an innovative method, the knowledge-guided multi-layer network (KGMN), to enable the global annotation of metabolites via the integration of three-layer networks, including (i) a knowledge-based metabolic reaction network, (ii) a knowledge-guided MS/MS similarity network, and (iii) a global peak correlation network [59].

Advanced data normalization tools have also been developed for metabolomics to render data comparable among different laboratories. Metabolite structure identification software is commonly employed for non-targeted metabolomics. For instance, Roberts et al. used Compound Discoverer software to demonstrate that deoxycytidine and ureidopropionate levels indirectly reflect SARS-CoV-2 viral load [39]. Alternative data normalization tools were developed as R packages to facilitate interexperimental comparisons.

#### 2.1.4. Statistical Analysis and Interpretation

Multivariate statistical methods are a critical part of the metabolomics workflow for the extraction of reliable information from complex data sets and the elimination of spurious correlations [60]. Standard tools for data analysis include principal component analysis (PCA), partial least square discriminant analysis (PLS-DA), and orthogonal partial least squares (OPLS), and all these tools have been used to distinguish the overall profile of uninfected and SARS-CoV-2-infected patients [43,44,61,62]. Functional analysis software is evolving to identify alterations in metabolic pathways informing on the severity of COVID-19 [42,43,63]. However, this latter approach does not contribute to the establishment of a cut-off for the identification of meaningful metabolic biomarkers. Other methods are based on the false discovery rate (FDR), and volcano-plots have been used to identify differentially abundant metabolites, for instance to discriminate healthy individuals from moderate or severe COVID-19 patients [63]. Robustness and accuracy can be estimated using ROC (receiver operating characteristic) analyses to assess metabolite concentrations [52,64]. ROC curves have been used to calculate the area under the curve (AUC) to identify the 10 most relevant metabolites for diagnosing COVID-19 with the best sensitivity and specificity [52,64]. Another statistical model, the logistic regression analysis, can be utilized to select the highest ranked metabolites which contribute the most to the discrimination among patient groups [64,65,66]. Random forest classifier and machine learning (ML) approaches can detect complex relationships among variables and can be employed successively on training and validation datasets [14,67]. Altogether, there is a panoply of different statistical methods that must be used in an adequate fashion for the analysis of complex datasets. Unfortunately, there is no fully standardized workflow for the statistical and bioinformatic treatment of metabolomics data that would facilitate the comparison of distinct studies.

## 3. Prognostic and Diagnostic Features of the Metabolome in COVID-19

In the preceding sections, [52,64], the wide variety of analytical methods and separation techniques (capillary electrophoresis or chromatography), as well as approaches (non-targeted, targeted), illustrates the diversity of results obtained by the different research groups [68]. In spite of the heterogeneity of methodologies, metabolomic analyses pinpoint profound effect of SARS-CoV-2 infection at the multiorgan level and more specifically at the level of the immune defense [2]. To interpret perturbations observed in COVID-19 metabolomics profiles, we reviewed the relevant PubMed-accessible literature with the aim to identify candidate biomarkers that should (i) correctly identify the presence of the COVID-19, (ii) predict or detect clinical progression, (iii) preemptively identify individuals at high risk of disease progression, and possibly (iv) generate information on the pathogenesis.

### 3.1. Method

We searched the PubMed database for articles up to December 2022. All articles were considered potentially useful if they covered the topics of COVID-19 and MS-metabolomic approaches in plasma/serum or other biological samples. This search initially yielded 118 publications, some of which were excluded because they employed other methodological approaches (e.g., lipidomic approach, nuclear magnetic resonance) or were not accessible as full texts. Eligible studies included male and female COVID-19 patients with a positive RT-PCR test for SARS-CoV-2, as well as healthy volunteers with a negative RT-PCR test. In order to focus on COVID-19 focus, we did not consider studies centered on participants with other initial diseases, such as lung diseases, cancer, inflammatory bowel disease, or patients who received therapeutic agents/vaccines. All articles were screened for metabolomic biomarkers that might reflect COVID-19 severity. References cited by the most relevant studies were scrutinized to identify additional publications.

As a result, 20 articles were included in this review. For each among these studies, we carefully compared the clinical design as well as the metabolomic profiling techniques (biological specimen, metabolites isolated, annotated biochemical pathways), as summarized in Table 1.

### 3.2. Clinical Significance of Prognosis Circulating Metabolome in COVID-19 Patients

A pioneering study by Wu et al. 2020 at Wuhan Jinyintan Hospital, reported targeted metabolomic profiles of the plasma collected from patients with COVID-19. The authors discovered five plasma metabolites (malate, aspartate, D-xylulose-5-phosphate, guanosine monophosphate (GMP), carbamoyl phosphate) to be downregulated in severe COVID-19 [43]. Subsequently, many other studies reported the identification of various potential candidate biomarkers for COVID-19 (Table 1). 

For instance, a strong alteration in amino acid metabolism compared to healthy volunteers was detected in the catabolic pathways affecting arginine, glutamine, branched-chain amino-acids and their derivatives (tryptophan, proline, lactate, glycine, phenylalanine, tyrosine, aspartate). These key signatures are sustained in mild/moderate and severe COVID-19 patients (Table 1). Among these, glutamate was found to be dysregulated in longitudinal studies across various COVID-19 waves [44]. Tryptophan metabolism was also confirmed to be perturbed in several studies, and two immunosuppressive tryptophan metabolites, kynurenine and anthranilic acid, were found to be associated with disease severity [41,67,69]. Elevations in such tryptophan metabolites were also found in patients with long-COVID or with cancer [37,71]. It is interesting to note that elevations of kynurenate accompany clinical deterioration in male (but not female) COVID-19 patients [66]. α-hydroxylated amino acid increased with disease severity and was related to reduced oxygen saturation and clinical markers of lung damage [18,61,64,69]. In addition to these amino acid-focused studies, alterations in carbohydrate and energy metabolism, tricarboxylic acid cycle (TCA), purine metabolism (adenine, GMP, cysteine, urea), polyamines, and nicotinamide metabolites have been reported. Thus, multiple glycerophospholipids, including phosphatidylethanolamines (PEs), lysophosphatidylethanolamines (LysoPE), sphingolipids, ceramides (Cer), and triglycerides (TG) or palmitic acid, were found to be altered in samples from severe COVID-19 patients [21,69,72]. Sphingosine-1-phosphate significantly increases during the recovery process [21,73]. Bilirubin and its degradation products, as well as specific bile acid derivatives, may reflect hepatic damage during COVID-19 disease progression [21,54]. Moreover, elevated creatine, and acetylated polyamines likely reflect renal dysfunction in severely ill COVID-19 patients [41,42] (Table 1). A recent meta-analysis of COVID-19-relevant circulating metabolomes has pooled the data from 272 COVID-19 infected-subjects and 120 healthy controls, revealing that the major biomarkers are cholesterol, D-mannose, tyrosine, L-phenylalanine and bilirubin [73]. This work further suggests that the severity of COVID-19 disease is associated with perturbed metabolic pathways involving phenylalanine, tyrosine, and tryptophan, as well as changes in the abundance of specific metabolites (L-alanine, uridine and uracile). Nevertheless, such meta-metabolomic studies are still hampered by differences in extraction procedures and analytical platforms.

### 3.3. Other Metabolomes

The untargeted metabolomic analysis of saliva and blood samples from 43 non-COVID-19 patients and 40 non-severe COVID-19 patients by Spick et al. led to the identification of two circulating molecules predictive of severity, glycolithocholic acid 3-sulfate and L-proline betaine, as well as the detection of an increase in salivary LPC aC18:2, Sarcosine C5-DC (C6-OH), SM C24:1, and trans-4-hydroxyproline [48]. Frampas et al. found that the concentration of valine, leucine, phenylalanine, tyrosine, and proline in saliva allowed to discriminate patients with mild and severe COVID-19 [74]. One study of the exhaled air metabolome failed to discover metabolites that correlate with COVID-19 disease severity and viral SARS-CoV-2 load [49]. In contrast, Barberis et al. reported that the abundance of fatty acids (1-monomyristin and monolaurin) in exhaled breath condensate can be used to discriminate COVID-19 patients from healthy controls and patients with other respiratory diseases [53].

SARS-CoV-2 may influence the nasal metabolome. Metabolomic analysis of nasopharyngeal swabs from mild COVID-19 patients has resulted in the detection of specific analytes, including LPCs C18:2, beta-hydroxybutyric acid, methionine sulfoxide, and carnosine, compared to other respiratory virus, such as influenza or respiratory syncytial virus [18]. Several metabolites (cyclohexanecarboxylic acid, lactate and urea) were found increased, but others (1-pentadecanol, D-cellobiose, deoxycholic acid, monomethyl succinate and propanoic acid) were depleted in fecal samples from patients with severe COVID-19 compared to mild disease [75]. Another stool metabolomic analysis correlated impaired tryptophan metabolism with a decreased abundance of the microbial metabolite indole-3-propionic acid, which is also decreased in serum of critical COVID-19 patients [76]. In the stool from COVID-19 patients, amino acids (glutamine, threonine, proline, glycine, tryptophan, phenylalanine, tyrosine, aspartic acid, leucine, and valine), as well as spermidine, putrescine, and vitamin B6, were found to be increased [77].

All these local or systemic metabolomes established a set of candidate metabolites serving a potential biomarker for COVID-19 progression and prognosis. Of note, some specific metabolome pathways were not compartment-specific. As an example, branched-chain amino acids (BCAA) were found to increase in both the circulating and fecal metabolomes [21,77]. Nonetheless, most of the candidate biomarkers were increased or decreased in an organ-specific fashion.

### 3.4. Diagnostic Metabolites Predicting the Progression of the COVID-19

The key goal of metabolomics is to quantitatively and qualitatively evaluate metabolites for their diagnostic potential [78]. As defined by the Food Drug Administration (FDA), a diagnostic biomarker detects or confirms the presence of disease, condition of interest, or identifies an individual with a subtype of the disease [79]. Compared to RT-PCR performed on nasopharyngeal swaps, diagnostic tests based on the detection of SARS-CoV-2 antigen are known to be rather limited in their sensibility [80,81]. Based on the current literature (Table 1), it appears that an increase in L-cytosine correlates with SARS-CoV-2 infection and hence might be considered as a diagnostic biomarker [39,52]. It has been hypothesized that increases of L-cytosine levels are critically involved in the evolution of RNA viruses, including SARS-CoV-2 [17]. Indeed, the underrepresentation of cytosine in the SARS-CoV-2 genome suggests a role other than viral RNA synthesis [82]. Furthermore, another study described viral replication to be particularly dependent on extracellular carbon sources such as glutamine [83]. Thus, the mechanistic roots of the elevation of L-cytosine correlating with SARS-CoV-2 infection are still unclear.

Metabolomics analyses methods were used to discriminate acute respiratory distress syndrome (ARDS) associated with COVID-19 from non-COVID-19 ARDS cases, based on the VOC compounds, including methylpent-2-enal, 2,4-octadiene, 1-chloroheptane, and nonanal, in patients’ exhaled air [49]. Metabolomic analysis associated increasing disease severity (mild, moderate, severe, critical, fatal) with the elevation of anthranilic acid, a kynurenine metabolite [41]. An increase in the levels of other tryptophan metabolites, such as kynurenic acid and kynurenine, supports the role of this pathway in disease aggravation [36,67]. Lewis et al. identified an elevation of circulating TG (22:1_32:5), TG (18:0_36:3), and glutamic acid during the different waves of the COVID-19 pandemic [44]. Delafiory et al. combined metabolomics and machine learning to diagnose COVID-19 by measuring 19 metabolites, including increased guanosine, uridine, deoxyguanosine, N-linoleoyl-glycine, specific N-acylethanolamines, PG, and PE, relating them to the pathophysiology of the disease [14]. Attempts have been launched to complement standard microbiological and biochemical methods of infection diagnosis by metabolomic analyses to guide specific and tailored antimicrobial therapies for critically ill COVID-19 patients [84]. In addition, urine-based diagnostic tests have been proposed as “comfortable” by Moura et al. to increase the adherence of individuals to routine testing [85]. The authors developed a rapid procedure based on multiplex flow injection analysis using tandem mass spectrometry (FIA-MS/MS). In urine, 14 molecules, including glycine, valine, glutamate, and tryptophan, allowed to diagnose COVID-19 with high sensitivity (>90%), specificity (>95%), and accuracy (>95%), which would be better than antigen detection in oropharyngeal swabs (32–48% positivity) and nasopharyngeal swabs (63% positivity) [86,87,88,89,90]. However, this urinary test would rely on costly MS-metabolomic equipment and has not been validated yet.

Indeed, as a common leitmotif, large-scale validation of metabolomic approaches for COVID-19 diagnosis is still pending, meaning that it is elusive which among the possible specimens (exhaled air, plasma, saliva or urine) would be optimally suitable for detecting and staging COVID-19.

## 4. Mechanistic Biomarkers of COVID-19

We wondered whether some of the candidate biomarkers metabolites might constitute ‘mechanistic’ biomarkers, which would be ‘actionable’ because they are causally involved in COVID-19 disease pathogenesis. Such mechanistic biomarkers would directly contribute to excessive inflammatory reactions and deficient immune control of SARS-CoV-2 (Figure 2). 

### 4.1. Tryptophan

Clinical studies suggested that tryptophan catabolism (i.e., reduced tryptophan and augmented levels of kynurenine, kynurenate and anthranilic acid) is associated with the severity and the progression COVID-19 [91]. These data are reinforced by transcriptomic data [92]. The main route of tryptophan catabolism is the kynurenine pathway, with the rate-limiting enzymes being tryptophan-2, 3-dioxygenase (TDO) and indoleamine-2, 3- dioxygenase 1/2 (IDO 1/2). These enzymes metabolize tryptophan to kynurenine and nicotinamide adenine dinucleotide (NAD^+^) [93,94]. Increased kynurenine levels correlate factor α (TNF-α) [92]. These latter factors (IFN-γ and TNF-α) have been suggested to activate IDO, hence closing a vicious feedforward loop [95]. In line with this hypothesis, epacadostat, an inhibitor of IDO1, reduces the release of proinflammatory cytokines by with adverse clinical outcomes and inflammatory properties, including elevated interleukin-1α and β (IL-1α, IL-1β), interleukin-6 (IL-6), interferon-γ (IFN-γ), and tumor necrosis circulating leukocytes from SARS-CoV-2-infected macaques [71]. Kynurenine can signal through the aryl hydrocarbon receptor (AHR). IDO1-dependent induction of AHR signaling by SARS-CoV-2 leads to the upregulation of downstream effectors and enhanced cytokine expression (e.g., IL-1β, IL-10, and TNF-α), hence further stimulating inflammation [96]. Besides kynurenine, the mechanistic contribution of anthranilic acid has been investigated in the pathogenesis of COVID-19. Anthranilic acid is best known as a third immediate downstream product of kynurenine. Anthranilic acid concentrations in the plasma robustly correlate with circulating interleukin-10 and -18 levels [41]. Moreover, 3-hydroxy-anthranilic acid was found to be accumulated in the brain, heart, and lung from COVID-19 patients, but this accumulation, which was detected on autopsy material, was correlated with the expression of the IDO2 isoform, not IDO1 [97]. Interestingly, anthranilic acid has been also associated with the risk of developing severe dengue fever, suggesting that this metabolite might be endowed with broad pro-inflammatory properties [98]. Indeed, high plasma levels of anthranilic acid correlate with the accelerated development of cardiovascular disease in patients with chronic kidney disease [99].

### 4.2. Other Amino Acids and Derivatives

Many amino acids and their derivatives are modified by COVID-19 and are associated with the severity of the disease, as reported for valine, L-citrulline, L-isoleucine, asparagine, aspartate, arginine, proline, and glycine [43,61,67]. Glutamine plays biosynthetic and bioenergetic roles in several pathways, including the TCA cycle, nucleotide, and fatty acid biosynthesis. It also acts as a fuel for immune cells, including macrophages and lymphocytes [100]. COVID-19 apparently boosts its degradation, as well as an elevation of glutamate [101]. A recent report provided some evidence linking comorbidity-associated glutamine deficiency with predisposition to severe COVID-19 [102]. Glutamate negatively correlates with the pro-inflammatory cytokines IL-6 and IL-18 but positively correlates with (especially CD4^+^) T cells [44,63,95]. Of note, nutritional glutamine supplementation caused a decrease of IL-1β, TNF-α and high-sensitivity C-reactive protein (hs-CRP), in COVID-19 patient sera, compared to the non-supplemented control group [103]. These finding establish the role of glutamine depletion in the pathogenesis of COVID-19. As mentioned above, BCAA catabolism, affecting valine, leucine, and isoleucine, is detected in mild, moderate, and severe/critical SARS-CoV-2 infection, as indicted by a decrease of circulating valine and isoleucine [52,61,65,70]. BCAA are essential for skeletal muscle and whole-human body anabolism and energy homeostasis [104]. Funneling of BCAAs from skeletal muscle appears to be an essential compensatory mechanism to hypoxia response, to regenerate NAD and NADP, as demonstrated for infections by *Aspergillus nidulans* and *Klebsiella pneumoniae*. Surprisingly, high levels of BCAA in fecal materials have been associated with elevated levels of interferon γ, interferon λ3, interleukin 6, CXCL-9, and CXCL-10 [78,105]. These studies suggest, but do not prove, that the fecal microbiota modulates the inflammatory tonus via metabolic effects.

COVID-19 pneumonia is associated with reduced bioavailability and low arginine-to-ornithine ratio in the plasma [106]. Accordingly, arginase 1 is upregulated in peripheral blood mononuclear cells (PBMCs) from COVID-19 patients [107]. Arginine plays a vital role in the immune response, and as a precursor of nitric oxide (NO) that is produced by NO synthase (NOS) [108]. Altered arginine metabolism is may reflect the endothelial dysfunction observed in COVID-19 pneumonia [106]. A clinical study in Beijing, conducted during the SARS-CoV-1 outbreak in 2003, found that NO inhalation therapy in critically ill SARS patients resulted in improved arterial oxygenation and provided noninvasive pressure support [109]. Based on these results, it has hypothesized that (i) NO directly inhibits the replication of SARS-CoV, and that (ii) NO exerts immunostimulatory effects on T lymphocytes. Indeed, in vitro experiments have demonstrated that the proliferative capacity of T-cells is significantly reduced in COVID-19 patients and can be restored by arginine supplementation [110]. Based on these arguments, a randomized clinical trial has evaluated the effects of oral arginine supplementation on patients hospitalized with COVID-19 as an add-on to standard of care. Patients treated with L-arginine exhibited a significantly reduced reduction of respiratory assistance during convalescence and a shorter hospitalization as compared to placebo-treated controls [111]. Hence, a relative deficit in arginine is likely to contribute to the pathogenesis of COVID-19.

### 4.3. Polyamines

Polyamines are critically involved in the maintenance of cellular homeostasis [112]. Polyamines, including putrescine, spermidine, spermine, and their derivatives, also emerged as important modulators of virus–host interactions [113]. A set of metabolomic studies demonstrated that circulating polyamines and their precursor L-ornithine are overabundant in sera of COVID-19 patients. In particular, acetylated polyamine derivatives (N1-acetylputrescine, N1-acetylspermidine, N1,N8-diacetylspermidine and N1,N12-diacetylspermine) were found to be elevated in COVID-19 samples as compared to controls [37,39,64]. Interestingly, serum levels of N1-acetylspermidine and N8-acetylspermidine are particularly high in long-term SARS-CoV-2 carriers [37]. Recent preclinical work indicates that SARS-CoV-2 infection causes the accumulation of key metabolites in infected cells, including the deregulation of putrescine and N-acetylspermidine by hijacking autophagy to facilitate their own viral replication [114]. In vitro experiments have shown that spermidine and spermine can facilitate SARS-CoV-2 infection and replication, especially when ornithine decarboxylase I is inhibited [115]. However, it has also been reported that spermidine and spermine inhibit SARS-CoV-2 infection via the induction of autophagy [114]. The levels of circulating N_1_-acetylputrescine correlate with IFNα2a, and IFNγ, IL-2, and IL-10 [37]. Goubet et al. demonstrated that N_1_, N_8_-diacetylspermidine anticorrelated with absolute lymphocyte counts and increased in the serum of cancer patients that failed to control SARS-CoV-2 infection of the nasopharynx as indicated by prolonged viral shedding (>40 days) [47]. It should be noted that, in contrast to spermidine, which has anti-inflammatory and immunostimulatory effects [116,117,118], acetylated polyamines have no such effects and may actually reflect spermidine catabolism secondary to the activation of spermidine/spermine N1-acetyltransferase-1 (SAT1) by type-1 interferons [113]. However, this conjecture requires further in-depth investigation.

### 4.4. Fatty Acids: Case of Palmitic Acid and Arachidonic Acid

Alterations in plasma FAs in adults with COVID-19 have been observed [119]. Severity of COVID-19 correlates with an elevation of arachidonic acid (AA) [66], an unsaturated ω6 fatty acid constituent of phospholipids, as well as a precursor of prostaglandins and thromboxanes (which are both pro-inflammatory), hydroxyeicosatetraenoic acids (HETEs), and epoxyeicosatrienoic acids (EETs) (which are endowed in anti-inflammatory properties) [120]. The cytokine storm of COVID-19 patients (with an elevation of IL-1β, IL-6 and TNF-α) has been related to the increase of eicosanoids including AA and its metabolites [121].

Palmitic Acid (PA), a long chain saturated fatty acid, may also contribute to the progression of COVID-19. PA, which is significantly increased in patients with COVID-19, might play a pro-inflammatory role [54,70]. PA elicits inflammatory signaling by macrophages and triggers chronic vascular inflammation in vivo and in vitro [59]. Moreover, PA is required for the palmitoylation of the SARS-CoV-2 spike protein, which is essential for viral infectivity [122]. In a preclinical mouse model of SARS-CoV-2 infection, injections of orlistat, a powerful inhibitor of fatty acid synthase, reduced SARS-CoV-2 replication in the lung, attenuated the pneumonia, and increased animal survival [16]. Altogether, it appears that PA and AA are causally involved in the processes leading to excessive COVID-19- associated inflammation.

### 4.5. Sphingolipids: Ceramides and Shingosine-1-Phosphate

The most frequently reported sphingolipids metabolites associated to COVID-19 disease are ceramide (Cer) and sphingosine-1-phosphate (S1P), which are known for their involvement in immune and inflammatory disorders [21,62,123]. High plasma Cer levels, including Cer (d18:1/16:0) and Cer (d18:1/24:1), are strongly associated to clinical COVID-19 severity [62,124]. Previous studies have implicated Cer in lung inflammation [125]. The decrease of circulating S1P appears to be associated to the activation of macrophage and their recruitment to sites of inflammation [12,21,42,65,126], as well as the COVID-19-related cytokine storm [127]. Intracellular S1P levels in erythrocytes are increased in COVID-19 patients compared with healthy controls due to upregulation of the S1P producing sphingosine kinase 1 and downregulation of the S1P degrading lyase [128]. S1P analogs have been proposed as a therapeutical strategy to protect from the cytokine storm induced by influenza virus H1N1 and respiratory syncytial paramyxovirus [129,130]. Hence, both S1P and Cer emerge as targetable disease-relevant factors.

### 4.6. Vitamin B3: Trigonelline and Nicotinamide

Trigonelline, which is formed by the methylation of the nitrogen atom of Nicotinamide (also called niacin or vitamin B3) diminishes with COVID-19 disease severity [41,63]. Trigonelline is a natural alkaloid, attenuating pro-inflammatory cytokines production, such as TNF-α and IL-6 in lung and spleen [131,132,133]. Nicotinamide and nicotinamide mononucleotide (NMN), the precursor of the coenzymes NAD^+^ and NADP^+^, are also depleted in COVID-19 [36,52,71]. Mehmel et al., suggested that adaptive immune response and overexpression of CD38 in both CD4^+^ and CD8^+^ lymphocytes correlate with lack of NAD^+^, leading to increased proinflammatory cytokines [134]. Xiao et al. described that altered vitamin B metabolism correlated with IL-6, IL-10, and IL-15 in serum samples from COVID-19 patients [71]. It has been speculated that nicotinamide riboside directly inhibits SARS-CoV-2 entry, replication, and transcription [135]. However, nicotinamide supplementation has failed to improved lymphopenia in COVID-19 patients, arguing against the conjecture that vitamin B3 deficiency would contribute to disease pathogenesis [136].

### 4.7. 3-Hydroxybutyric Acid

In some studies, 3-hydroxybutyric acid (3HB), a ketone body produced by the liver upon fasting, prolonged exercise, and carbohydrate restriction, was found to be elevated in the course of COVID-19 [18,54,64]. Moreover, 3HB directly inhibits the inflammasome [137] and has immunostimulatory properties, increasing the production of IFN-γ by CD4^+^ T-cells and the cytotoxic activity of CD8^+^ T cells [138]. In mouse models 3HB improved the function of CD4^+^ T-cells and reduced the mortality in SARS-CoV-2 infected mice [139]. Hence, 3HB may act as a disease-attenuating metabolite in COVID-19.

In this section, we discussed a comprehensive approach that reflects COVID-19 disease process activity and identifies metabolites that are (positively or negatively) related to disease progression and its immunopathological correlates. We identified eight classes of biomarkers, including tryptophan metabolites, specific amino acids, vitamin B3, 3-hydroxybutyric acid, ceramides and shingosine-1-phosphate, palmitic acid, arachidonic acid, and polyamines (Figure 2). Pending future validation, these eight classes of metabolites can be used as potential diagnostic, prognostic, and mechanistic biomarkers.

## 5. Conclusions

Over the past decades, metabolomic approaches have continuously been refined, thus allowing for ever more complete coverage of the metabolome. Despite this technological progress, standardization of the entire methodological workflow remains a major issue that must be addressed to render metabolomics compatible with clinical routine. In spite of the heterogeneity of methods that renders comparison among studies difficult, we identified a COVID-19-relevant catalogue of potential diagnostic and prognostic biomarkers. These metabolites are more than mere biomarkers and are likely to contribute to excessive inflammatory reactions and mitigate the deficient immune control of SARS-CoV-2. Future studies should determine which prognostic biomarkers, alone or in aggregate, in exhaled air, plasma, saliva, or urine, most accurately reflect SARS-CoV-2 infection, COVID-19 severity, and adequate patient management.

## Figures and Tables

**Figure 1 metabolites-13-00342-f001:**
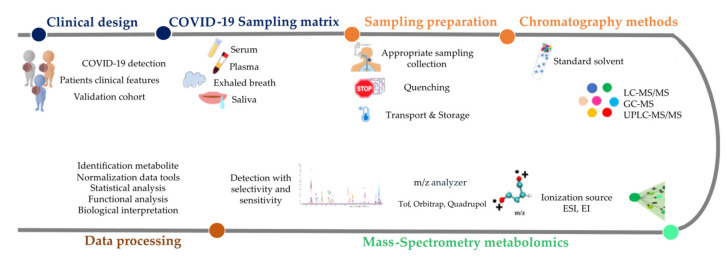
Metabolomics workflow to guideline towards a prognostic biomarker of COVID-19 identified by mass spectrometry metabolomics. Abbreviations: SARS-CoV-2: Severe Acute Respiratory Syndrome Coronavirus-2; COVID-19: Coronavirus disease 2019.

**Figure 2 metabolites-13-00342-f002:**
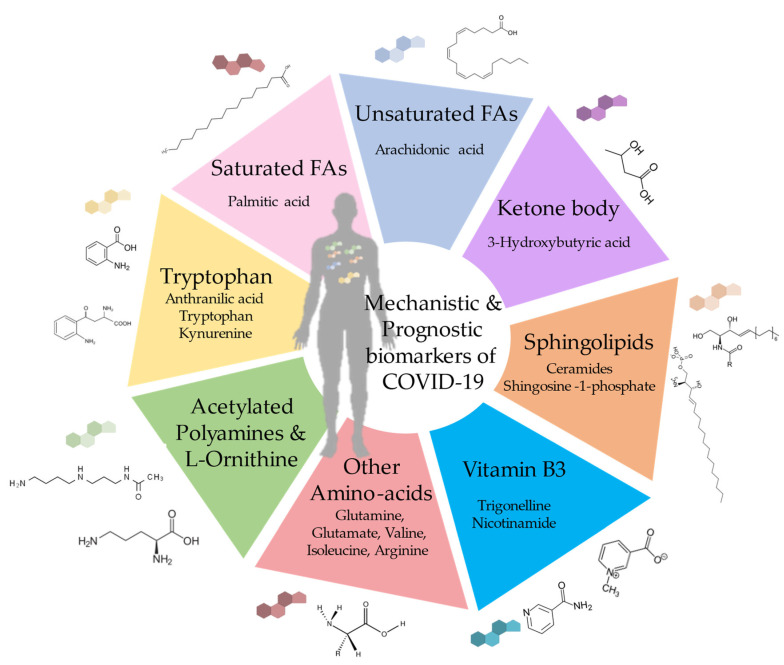
Mechanistic, diagnostic and prognostic biomarkers of COVID-19 disease.

**Table 1 metabolites-13-00342-t001:** Metabolite prognostic and diagnosis associated with COVID-19 disease using metabolomic MS-techniques.

Authors	Biological Matrix	COVID-19 Infected Patients	MS Techniques	Statistics & Data Normalization	Number of Metabolites Detected	Pathway Associated to COVID-19	Top of the COVID-19 Metabolites Biomarkers	Robustness, Originality, and Limits
Wu et al., 2020 [43]	Plasma	9 fatal outcomes, 11 severe, 14 mild, 10 Healthy subjects	LC-ESI-MS/MS	OPLS-DA, functional enrichment analysis, logistic regression analysis	431 metabolites common for all COVID-19 patients	pyrimidine, urea cycle, fructose and mannose, carbon	(↓) malic acid (↓) aspartic acid, (↓) D-xylulose 5 phosphate, (↓) guanosine monophosphate (GMP), (↓) carbamoyl phosphate	Longitudinal studies, associations of age and gender with COVID-19, low sample size inpatients
Barberis et al., 2020 [69]	Plasma	COVID-19 patients (*n* = 103), non-COVID-19 with symptoms (*n* = 32) and healthy controls (*n* = 26)	LC-MS/MS	PCA, volcano plots, MSE analysis, ROC analysis	75 modulated metabolites	phenylalanine, tyrosine and tryptophan biosynthesis, phenylalanine metabolism, aminoacyl-tRNA degradation, arachidonic acid metabolism	(↑)2-hydroxy-3-methylbutyric acid, (↑) 3-hydroxyisovaleric acid, (↑) 2-hydroxybutyric acid, (↑)palmitic acid, (↑)pyroglutamic acid, (↓)L-valine	Large number of patients (*n* = 161) but absence of asymptomatic COVID-19 patients
Song et al., 2020 [65]	Plasma	Controls (*n* = 26), mild COVID-19 (*n* = 18), moderate (*n* = 19), critical (*n* = 13) patients	UPLC-MS/MS	Logistic regression model with leave-one-out (LOO) cross-validation	404 metabolites	β-oxidation, TCA cycle, steroid pathway, amino acids	(↓) sphingosine-1-phosphate, (↑) biliverdin, (↑)5-hydroxy-tryptophan, (↓)tryptophan (↓)valine, (↓) proline, (↓)citrulline	Quantitative serum lipidome and metabolome but small longitudinal cohort
Thomas et al., 2020 [42]	Sera	33 COVID-19–positive and 16 control COVID-19–negative	UHPLC-MS	One-way ANOVA with Tukey’s multiple comparisons Spearman’s correlations	206 targeted metabolites and 5518 untargeted metabolites	Nitrogen (amino acid homeostasis), carbon (glucose and free fatty acids), tryptophan/kynurenine pathway, oxidant stress (methionine sulfoxide, cystine), renal dysfunction (creatine, creatinine, polyamines).	_	Comprehensive serum metabolome with detailed metabolic pathway but a low number of samples
Cai et al., 2020 [66]	Sera	COVID-19 patients (*n* = 39) and uninfected controls (*n* = 20)	UPLC-MS/MS	Multivariable logistic regression, Spearman correlation analysis, Chord diagram	75 metabolites with 17 metabolites associated with COVID-19 status for age, BMI, sex, and multiple comparisons	Tryptophan pathway metabolites	(↑) kynurenic acid, (↓) glutamate, (↑) cysteine-S-sulfate, (↑) palmitoleic acid, (↑) arachidonic acid, (↑) lysophosphatidylethanolamine (LPE) (22:6), (↓) glutamine, (↓) tryptophan	Metabolites correlate with immune response in a sex-specific manner
Blasco et al., 2020 [52]	Plasma	55 patients infected with SARS-CoV-2 at the time of viral diagnosis (D0) and 45 controls	LC-HRMS	PCA, volcano plots, MSE analysis, Venn diagram, ROC curves	160 metabolites retained in the final dataset.	Nicotinate and nicotinamide metabolism, Arginine, proline and purine metabolisms	(↑) cytosine, (↑) indole-3-acetic acid, (↑) L-isoleucine, (↑) L-asparagine, (↑) 1-aminocyclopropanecarboxylate	Multivariable analysis. Discriminant metabolic pathways predict clinical outcomes of COVID-19 patients
Shen et al., 2020 [21]	Sera	28 healthy subjects, 25 non-COVID-19, 25 non-severe COVID-19, 37 non-severe and 25 severe COVID-19 patients	UPLC-MS/MS	Random forest machine learning model based on metabolomic data from 18 non-severe and 13 severe patients	From 941 metabolites identified, 204 metabolites at the final data set	Bilirubine products, tryptophan, glycerophospholipid, sphigolipids and fatty acids and amino acid metabolism	(↑) kynurenine, (↓) choline, (↑) mannose, (↓) serotonine, (↓) bilirubin degradation product	Hydrophilic and hydrophobic molecules and viable diagnostic and therapeutic tools, but sera samples collected at different time points
Caterino et al., 2021 [63]	Sera	9 healthy control and 52 hospitalized COVID-19 patients, mild (*n* = 20), moderate (*n* = 16), and severe (*n* = 16)	LC-MS/MS	PLS-DA volcano plots, Spearman correlation, MSEA	143 quantified metabolites	Glycolysis/Gluconeogenesis, D-glutamine and D-glutamate metabolism, nitrogen metabolism, arachidonic acid metabolism, amino acid metabolism	(↑) lactate (↑)glutamate, (↑)glycine, (↑)aspartate, (↓)trigonelline (↓) phenylalanine, (↓) arachidonic acid	Correlation with inflammatory cytokines (succinic acid, xanthine, ornithine and glutamate)
Delafiori et al., 2021 [14]	Plasma	350 controls, 442 COVID-19 confirmed and 23 suspicious patients	HESI-Q Exactive Orbitrap-MS	Machine learning	19 discriminant biomarkers for COVID-19 selected by the ML	_	(↑) guanosine, (↑) uridine, (↑) deoxyguanosine, (↑) N-linoleoyl-glycine, (↑) N-acylethanolamines (C20:1 and C22:0), (↑) phosphatidylglycerol (PG) [PG (20:5)], (↑) phosphatidylethanolamine (PE) [PE (38:4)], (↑) phosphatidylcholine (PC) [PC (38:8)]	COVID-19 automated diagnosis and risk assessment through metabolomics and machine learning
Khodadoust et al., 2021 [62]	Plasma	Active COVID-19-infected participants, including 18 severe respiratory distress and 32 with mild symptoms	UPLC − QTOF/MS	PCA OPLS-DA MEDM	283 lipids covering 8 lipid classes	PS, PEs, Cer, HexCer, Hex2Cer, and Hex3Cer, salvage of sphingosine, sphingolipids with sphingomyelin	(↑) Cer (d18:1/16:0) (↑) Cer(d18:1/24:1) subclasses	Interface between metabolomics and lipidomic for the identification of lipid metabolites
Danlos et al., 2021 [41]	Sera	Controls (*n* = 29), mild COVID-19 patients (*n* = 23), moderate cases (*n* = 21), critical patients (*n* = 28)	GC-MS UHPLC-MS/MS	PCA Wilcoxon rank-sum test random forest machine learning model	757 metabolites	_	(↑) anthranilic acid, (↑) 3-hydroxy-DL-kynurenine, (↑) 5-hydroxy-DL-(↓) tryptophan, (↓) desaminotyrosine, (↓)arginine, (↑) ornithine, (↑)spermine, (↑)spermidine	Correlations between cytokines and metabolites and anthranilic acid as a prognostic biomarker
Xiao et al., 2021 [70]	Sera	14 mild, and 23 severe COVID-19 patients and 17 healthy controls	UHPLC-MS/MS	Volcano plots	253 metabolites from 134 metabolites with targeted method and 155 metabolites identified from 6072 metabolites with untargeted methods	arginine metabolism, tryptophan, purine metabolism, nicotinate and nicotinamide metabolism, TCA cycle	_	Longitudinal metabolite–cytokine correlation in follow-up mild COVID-19 patients
Overmyer et al., 2021 [67]	Plasma	COVID-19 status and hospital-free days at day 45 with COVID-19 patients (*n* = 102) and non-COVID-19 patients (*n* = 26)	GC-MS analysis and AEX-LC-MS/MS	PCA, linear regression log-likelihood tests machine learning approach	110 metabolites and 511 unidentified metabolites features	-	(↓) salicylic acid, (↓) methylphenol, (↑) kynurenine, (↑) quinolinic acid	Machine learning with multi-omics data and cross-ome correlation analysis
Páez-Franco et al., 2021 [61]	Plasma	COVID-19 severe patients (*n* = 46), mild patients (*n* = 19)	GC/MS	PLS-DA hierarchical cluster analysis	-	Valine and threonine catabolism	(↑) three α-hydroxyl acids of amino acid	Comprehensive serum metabolome
Shi et al., 2021 [64]	Sera	79 COVID-19 patients, 78 healthy controls and 30 COVID-19-like patients	GC/MS	One-way ANOVA followed by the student-Newman-Keuls, ROC	75 metabolites	_	(↑) butyric acid, (↑) 2-hydroxybutyric acid, (↑) L-glutamic acid, (↑) L-phenylalanine, (↑) L-serine, (↑) 3-hydroxybutyric acid	Correlation with clinical features but no asymptomatic SARS-CoV-2 infected troll sols. GC–MS is limited
Sindelar et al., 2021 [36]	Plasma	Longitudinal studies with 272 SARS-CoV-2 positive patients and 67 SARS-CoV-2 negative patient with 3, 7, 14, 28 and 84 days after the initial blood collection	LC/MS-MS	PCA and HCA visualizations, ML model	First putative identification of 707 metabolites with 92 unique metabolites	_	(↑) kynurenate, (↑) nicotinamide, (↑)creatinine, (↑)serine	Predicted metabolites confirmed by in vivo experimentation
Valdès et al., 2022 [54]	Plasma	Negative controls (*n* = 25), positive asymptomatic patients (*n* = 28); mild (*n* = 27); severe (*n* = 36); fatal outcome (*n* = 29)	HPLC–QTOF–MS	ANOVA, U test and PLS-DA MSE	After Post-processing, 203 metabolites	Carnitines, Ketone body, fatty acids, lysophosphatidylcholines/phosphatidylcholines, tryptophan, bile acids and purines	(↓) hippuric acid, (↑) 3-Hydroxyphenylacetic acid, (↑) urea, (↑) 3-hydroxybutyric acid, (↑) xanthine, (↑) alpha-linolenic acid	Longitudinal studies
Chen et al., 2022 [15]	Sera	20 COVID-19 patients and 20 healthy	UHPLC-MS/MS	PLS-DA MSEA Volcano plots	Out of the 714 metabolites identified, 203 change significantly in COVID-19 patients	Amino acids, fatty acids (long-chain fatty-acid), and glycerophospholipids, bilirubin	(↑) linolenate, (↑) choline, (↑) glycerol-3-phosphate, (↑) glycerophosphocholine, (↑) di-homolinoleate	Limited sample size
Lewis et al., 2022 [44]	Sera	Longitudinal studies with 41 negative and 123 SARS-CoV-2 positive patients (with 32 wave 1 and 91 wave 2)	LC-MS	PCA OPLS-DA machine learning model prediction	30 metabolites selected with highest VIP scores		(↑) TG (22:1_32:5), (↑)TG (18:0_36:3), (↑)glutamic acid, (↓) glycolithocholic acid, (↑)aspartic acid	Lack of healthy control subjects, no information on viral strains
Roberts et al., 2022 [39]	Sera	Discovery cohort with 120 COVID-19 patients and additional 90 COVID-19 patients validation cohort	UHPLC-MS/MS	Univariate and a multivariable bayesian logistic regression model, pathway enrichment analysis	935 metabolic features identified that Bayesian logistic regression with 20 metabolites with relevant biological functions	pyrimidine metabolites, tryptophan—kynurenine degradation, deoxycytidine and ureidopropionate	(↑) ureidopropionate, (↑) cytosine (↓) uracil, (↓) arginine, (↓) tryptophan, (↑) N1-acetylspermidine	Multiple predictor Bayesian logistic regression model but LC solvents and gradient elution do not allow to reliably measure long chain acyl carnitines

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
