# Peer review of "Diagnostic, Prognostic and Mechanistic Biomarkers of COVID-19 Identified by Mass Spectrometric Metabolomics"

_metabolites, 2023, doi:10.3390/metabo13030342_

Round 1

Reviewer 1 Report

Seite 1 – Zeile 47:

The authors cite Ref. 8 in line 41 and continue with Ref. 11-13 in line 47. Presumably, refs. 4 & 5 in line 42 will be the accurate references 9 and 10. This needs to be corrected.

In lines 47 and 48, the statement is made that metabolomics allows characterising the complex relationship between SARS-CoV-2 and host metabolism [Ref. 15]. In line 53, the "big heterogeneity in methods" is mentioned, e.g. serum vs plasma, storage (-80°C vs liquid nitrogen), contamination when using saliva, chromatographic separation of metabolites, targeted vs untargeted metabolomic approach, some specific metabolome pathways are not compartment-specific, as well as differences in swab kits. Due to the numerous weaknesses and changes, this should be discussed in detail and compressed to a manageable density so that physicians can set a therapeutic measure quickly (like it was done in point 4). In addition, effects in numerous metabolic pathways are mentioned. This should be presented in more detail. In this context, the advantage of this method over specific methods, i.e., PCR determination, antibody assays, inflammation parameters, price-performance ratio, etc., should also be compared.

What are the reasons for the metabolic changes? Is the cause the virus per se, is it the activity of the virus in the host, or are the various changes due to the immune defence? There are differences between different viruses, but there also seem to overlap. This should be worked out more clearly.  

Figure 1:

"Clinical design - SARS-COVID19 detection is mentioned. Was this abbreviation deliberately chosen, or should it be SARS-CoV-2 or COVID-19?

Page 4 - line 147:

The authors mention Dettmer et al. 2007 - reference (12) should be added here.

Line 185:

Robert et al. should read Roberts et al. 2022.

Page 5 - line 207:

Dr Wu and his team should be replaced by Wu et al. 2020 [44].

Line 212-216:

The authors mentioned branched-chain amino acids (tryptophan,... where the end of the parenthesis is missing. Furthermore, glutamine is mentioned several times. This should be revised.

Page 5 - Point 3.1 Clinical significance:

The authors mention numerous studies with a wide variety of markers. The heterogeneity seems to need to be clarified in this regard. The authors should explain why the differences occur.

Page 10 - line 278:

The abbreviation BCAA must be explained at the first mention.

Page 11 - line 337:

(Mann et al. 2021) is missing from the reference list - this citation needs to be added.

Page 12 - line 363:

"Polyamines are -critically ..." what is the hyphen for?

Line 381:

SARS-CoV2 should read SARS-CoV-2 to be congruent. The same applies to line 460.

Page 12 - line 389:

The authors cite a 2018 publication to explain "alterations in plasma fatty acids in adults with COVID-19" - this seems questionable.

Page 13 - line 410:

... involvement in in ... - the repetition should be removed.

Page 13 - 4.6 Vitamin B3

The authors mention vitamin B3 in their research, but no other vitamins or antioxidants, especially since it has been proven that COVID-19 causes the complete loss of antioxidant capacity (Antioxidants 2021, 10, 1341, https://doi.org/10.3390/antiox10091341)! This should be supplemented.

References:

Citations 3, 45, 47, 49, 56, and 81 are deficient and need to be improved accordingly, e.g. Ref. 56: Diagnosis of COVID-19 by analysis of breath with gas chromatography-ion mobility spectrometry - a feasibility study. EClinical Medicine 2020 29:100609 https://doi.org/10.1016/j.eclinm.2020.100609.

In Ref. 47, the names were written in block letters.

Author Response

We thank the reviewer for his/her comment, and we adjusted all points in the revised manuscript and in the attachment.

Reviewer 2 Report

The authors critically discuss the use of mass spectrometric methods to analyze metabolites associated with COVID-19 pathogenesis. These metabolites are grouped into three groups that can serve for diagnostic, prognostic, and mechanistic studies of COVID-19.

Major comments:
In section "2.1.3 Metabolome Analysis and MS-based Data Processing" or maybe a new section, please discuss the approaches for statistical analysis and the criteria that are used to determining which metabolites should be considered differentially abundant.

There are sections in the review that discuss prognostic and mechanistic metabolites in detail, but no section on diagnostic metabolites. However, they are present in Table 1.
Please add a new section and discuss diagnostic metabolites in detail.

Minor comments:
Line 73 There are many microorganisms associated with the human body, so it is better to talk about many microorganisms rather than one.
Figure 1. Please translate the word "analyzeur" into English.
Line 85 The word versus should be italicized.
Line 89 In the phrase "These criteria are that" the word "that" should be replaced with another word such as "so" or "such".
Line 147 The reference should be formatted.
Line 279 Please remove the opening parenthesis before the references.

Author Response

 We thank the reviewer for his/her comment, and we adjusted all points in the revised manuscript and in the attachement.Please see the attachment.

Reviewer 3 Report

Dear Authors,

In the work entitled: "Diagnostic, prognostic and mechanistic biomarkers of COVID-19 identified by mass spectrometric metabolomics", the authors attempt to review the literature on the metabolome associated with COVID-19. To highlight the outstanding issues regarding the interaction between 1) SARS-CoV-2 virus, 2) host cell metabolism and 3) extracellular metabolites, mechanistic evidence was collected on diagnostic and prognostic biomarkers in various tissues. An integrated picture of biomarkers that may be useful in the clinical treatment of COVID-19 is also presented. Nevertheless, the work has several shortcomings and defects that should be eliminated before submitting the work to print.

Detailed notes:

1.     The abstract should be structured: background, purpose of the work, briefly methodology, results and general conclusion/

2.     The introduction lacks a clear purpose of the work.

3.     In the introduction, there is no alternative research hypothesis to the null hypothesis and its verification in the further part of the work.

4.     There is no chapter "Methodology/Methodology of research" where the authors should provide the methods and sources of the searched works, which databases were used.

5.     The description of the results of the literature review should be better structured.

6.     Conclusions should be generalizing and summarizing, and at least one conclusion should be towards the future.

7.     The bibliography should include more recent works from 2021-2023.

Author Response

 We thank the reviewer for his/her comment, and we adjusted all points in the revised manuscript and in the attachement. Please see the attachment

Round 2

Reviewer 1 Report

Page 5 - line 205: The citation of Roberts et al. 2022 has not been changed - "Robert et al. 2022" is still used here. This needs to be improved.

In line 354, COVID-10 is mentioned. Presumably, the authors mean COVID-19 - this should be corrected.

Author Response

We thank the reviewer for his/her comment, and we adjusted alls points in the revised manuscript. Please see the attachment.

Reviewer 2 Report

The authors have substantially revised the manuscript in accordance with the reviewers' comments. There are no further significant comments on my part and I can recommend the manuscript for publication in the journal.

Author Response

We thank the reviewer for his/her comment. Please see the attachment.
